# Osteoarthritis-Induced Metabolic Alterations of Human Hip Chondrocytes

**DOI:** 10.3390/biomedicines10061349

**Published:** 2022-06-08

**Authors:** Annett Eitner, Simon Sparing, Felix C. Kohler, Sylvia Müller, Gunther O. Hofmann, Thomas Kamradt, Hans-Georg Schaible, Matthias Aurich

**Affiliations:** 1Department of Trauma, Hand and Reconstructive Surgery, Experimental Trauma Surgery, Jena University Hospital, Friedrich-Schiller-University Jena, 07747 Jena, Germany; simon.sparing@med.uni-jena.de (S.S.); felix.kohler@med.uni-jena.de (F.C.K.); gunther.hofmann@med.uni-jena.de (G.O.H.); 2Institute of Immunology, Jena University Hospital, Friedrich-Schiller-University Jena, 07743 Jena, Germany; sylvia.mueller@med.uni-jena.de (S.M.); thomas.kamradt@med.uni-jena.de (T.K.); 3BG Trauma Center Bergmannstrost, 06112 Halle (Saale), Germany; matthias.aurich@uk-halle.de; 4Institute of Physiology 1/Neurophysiology, Jena University Hospital, Friedrich-Schiller-University Jena, 07743 Jena, Germany; hans-georg.schaible@med.uni-jena.de; 5Department of Orthopaedics, Trauma and Reconstructive Surgery, Halle University Hospital, Martin-Luther-University Halle-Wittenberg, 06120 Halle (Saale), Germany

**Keywords:** osteoarthritis, metabolic alterations, chondrocytes, oxygen consumption rate, mitochondrial ATP production, opioid peptides

## Abstract

Osteoarthritis (OA) alters chondrocyte metabolism and mitochondrial biology. We explored whether OA and non-OA chondrocytes show persistent differences in metabolism and mitochondrial function and different responsiveness to cytokines and cAMP modulators. Hip chondrocytes from patients with OA or femoral neck fracture (non-OA) were stimulated with IL-1β, TNF, forskolin and opioid peptides. Mediators released from chondrocytes were measured, and mitochondrial functions and glycolysis were determined (Seahorse Analyzer). Unstimulated OA chondrocytes exhibited significantly higher release of IL-6, PGE_2_ and MMP1 and lower production of glycosaminoglycan than non-OA chondrocytes. Oxygen consumption rates (OCR) and mitochondrial ATP production were comparable in unstimulated non-OA and OA chondrocytes, although the non-mitochondrial OCR was higher in OA chondrocytes. Compared to OA chondrocytes, non-OA chondrocytes showed stronger responses to IL-1β/TNF stimulation, consisting of a larger decrease in mitochondrial ATP production and larger increases in non-mitochondrial OCR and NO production. Enhancement of cAMP by forskolin prevented IL-1β-induced mitochondrial dysfunction in OA chondrocytes but not in non-OA chondrocytes. Endogenous opioids, present in OA joints, influenced neither cytokine-induced mitochondrial dysfunction nor NO upregulation. Glycolysis was not different in non-OA and OA chondrocytes, independent of stimulation. OA induces persistent metabolic alterations, but the results suggest upregulation of cellular mechanisms protecting mitochondrial function in OA.

## 1. Introduction

Osteoarthritis (OA) is characterized by degradation of articular cartilage, low-grade inflammation of synovial tissue, and alterations in subchondral bone [1]. Chondrocytes are considered an important cellular mediator of OA pathogenesis, which can undergo phenotypic modulation in response to environmental changes provoked by mechanical injury, overload or inflammation [2].

In experimental studies, the mechanical compression of cartilage as well as the cytokines produced by synoviocytes, macrophages or chondrocytes can modify matrix metalloproteinase (MMP) gene expression, collagen and further cytokine production [3,4]. However, permanent metabolic alterations in chondrocytes induced by OA, without additional stimuli, are rarely studied, and data are often based just on different gene expression profile analyses. A direct comparison of metabolic processes in non-OA and OA-modified articular cells is important for the identification of crucial metabolic alterations and the development of disease-modifying therapies.

In addition to modulation of matrix protein production, cytokines and other pro-inflammatory mediators can induce mitochondrial dysfunction in OA chondrocytes [5,6]. Mitochondria play a key role in aging-related diseases and are important for the cellular function and survival of chondrocytes [6]. Reduced activity of mitochondrial complexes was reported in OA chondrocytes, which may increase cartilage degradation, oxidative stress, cytokine production and apoptosis [6]. Increased oxidative stress may cause inhibition of the mitochondrial respiratory chain and ATP production and may induce mitochondrial DNA (mtDNA) mutation [7]. Alterations in mitochondrial function induced by interleukin-(IL-)1β, tumor necrosis factor (TNF) and nitric oxide (NO) have been described either in normal or OA human chondrocytes [5,8,9], but direct comparisons are lacking.

IL-1β induced the release of NO, interleukin-6 (IL-6), prostaglandin E_2_ (PGE_2_) and MMP-3 and resulted in mitochondrial alterations in OA-chondrocytes [5]. Increased cyclic adenosine monophosphate (cAMP)-signaling, a modulator of NO production, attenuated IL-1β-induced mitochondrial dysfunction, probably by reducing inducible nitric oxide synthase (iNOS) activity or expression [5]. Whether the mitochondrial activity of normal chondrocytes is modulated by cAMP has not been explored.

In addition to cytokines and MMPs, diverse neuropeptides are present in the synovial fluid of OA patients and affect the catabolic and anabolic processes of chondrocytes since chondrocytes express the corresponding receptors [10]. The impact of neuropeptides is different in non-OA and OA chondrocytes [10]. Substance P increased cAMP levels and IL-6 release in non-OA chondrocytes, whereas in OA chondrocytes, substance P did not alter the levels of cAMP and IL-6 compared to unstimulated cells [10]. The ERK and AKT pathways can be activated by substance P and calcitonin gene-related peptide alpha (αCGRP) in non-OA chondrocytes, whereas in OA chondrocytes both neuropeptides can only activate the ERK pathway. The authors concluded that both neuropeptides induce pro-inflammatory and destructive effects through the activation of ERK signaling but also induce anti-inflammatory and anabolic effects by increased cAMP signaling [10]. Chondrocytes also express functional receptors for endogenous opioid neuropeptides [11]. Interestingly, activation of opioid receptors inhibits adenylyl cyclase activity, reducing the cAMP level in several cells [12,13], whereas in chondrocytes, the opioid dynorphin A increases the intracellular concentration of cAMP [14]. Thus, the activation of opioid receptors may mediate chondro-protective effects, but an upregulation of pro-inflammatory mediators by opioid peptides in chondrocytes was also reported [14,15]. The effect of neuropeptides on inflammation-induced metabolic alterations and mitochondrial dysfunction is unclear and should be further investigated. New research focuses on the improvement of the mitochondrial function of OA chondrocytes by exogenous drugs as potential OA treatments [16].

The aims of our study were (1) to compare directly the basal activity of metabolic processes and mitochondrial function in OA and non-OA chondrocytes, (2) to investigate whether the inflammatory mediators IL-1β and TNF affect the metabolic processes and mitochondrial function of OA and non-OA chondrocytes in a similar way, and (3) to evaluate whether substances influencing cAMP signaling affect mitochondrial function in both non-OA and OA chondrocytes. Chondrocytes were isolated from human hip joint cartilage and stimulated with IL-1β or TNF, and in combination with forskolin (a stimulator of adenylyl cyclase) or opioid neuropeptides. Parameters of the mitochondrial respiration chain, non-mitochondrial oxygen consumption and glycolysis, as well as the release of NO, IL-6, MMP1, PGE_2_ and glycosaminoglycan (GAG), were analyzed. We hypothesized that normal chondrocytes have higher mitochondrial ATP production and are less sensitive to inflammatory stimuli.

## 2. Materials and Methods

### 2.1. Reagents/Solutions

Human IL-1β and human TNF were purchased from PeproTech (Rocky Hill, NJ, USA) and dissolved in water. Forskolin was purchased from Tocris Bioscience (Bristol, UK) and dissolved in dimethyl sulfoxide (DMSO)/water (final DMSO dilution 1:200). Β-Endorphin, Nociceptin and Dynorphin A were purchased from Cayman Chemical (Ann Arbor, MI, USA) and dissolved in water. Pronase E was obtained from Merck KGaA (Darmstadt, Germany), and collagenase P from Roche Diagnostics GmbH (Mannheim, Germany). The chondrocyte culture medium contains Chondrocyte Basal Medium + 10% Chondrocyte Growth Medium SupplementMix (both from PromoCell GmbH, Heidelberg, Germany) + 1% penicillin/streptomycin solution (Life Technologies Europe BV, NN Bleiswijk, The Netherlands). For all stimulation experiments, stock solutions of cytokines, forskolin and opioid neuropeptides were diluted to the final concentration in chondrocyte culture medium.

### 2.2. Patient Material

Human hip chondrocytes were obtained from 24 patients (16 females/8 males) with end-stage hip OA (N = 14) or femoral neck fractures (N = 10) who underwent hip arthroplasty. The non-OA group included only patients with femoral neck fractures who did not have any problems or pain in the hip before fracture and were without any signs of osteoarthritic alterations of the cartilage. Patients were, on average, 70.17 years old (±7.9 years, standard deviation). The age of non-OA patients and OA-patients was not significantly different. Patients were informed about the purpose of tissue sampling and gave written consent. The study was approved by the Ethical Committee for Clinical Trials of the Friedrich Schiller University of Jena and of the Sächsische Landesärztekammer (5208-07/17; EK-BR-81/17-1) and performed in accordance with the Declaration of Helsinki.

### 2.3. Isolation of Human Hip Chondrocytes

Cartilage of the femoral head was removed and cut into small pieces. For isolation of chondrocytes, cartilage was treated with 0.01 mg/mL pronase E in Dulbecco’s modified Eagles’s medium (DMEM) for 30 min at 37 °C following collagenase P (1.3 mg/mL in chondrocyte culture medium) for 16 h at 37 °C. The cells were filtrated, washed and directly seeded in the appropriate cell culture plates for stimulation experiments.

### 2.4. Experiments on Release of Mediators

For release experiments, isolated chondrocytes were plated on 24-well culture plates at a density of 4 × 10^4^ cells/cm^2^ and cultured in chondrocyte culture medium. After 3 days of incubation, the medium was renewed. After additional 2 days, the cells were stimulated with IL-1β (0.1 ng/mL) or TNF (0.1 µg/mL), and in combination with 50 µM forskolin (all diluted in chondrocyte culture medium) for 48 h. Additionally, cells were stimulated with IL-1β or TNF in combination with β-endorphin (1 µg/mL), nociceptin (1 µg/mL) or dynorphin A (1 µg/mL) for 48 h. Effective concentrations of all used mediators were determined in preliminary experiments (see Appendix A). Supernatant of all experiments was collected and stored at −80 °C until analysis. The cells were plated in duplicate for each condition. Experiments were performed with a minimum of 5 biological replicates (donors) to ensure reproducibility.

Griess assay: The concentration of nitrite in the supernatant was measured using the Griess Reagent Kit (#G7921, Invitrogen, Thermo Fisher Scientific Inc., Darmstadt Germany), according to manufacturer’s instruction and used as an indicator for NO synthesis of the cultured chondrocytes. The absorption was measured at 548 nm using the microplate spectrophotometer Epoch and analyzed using the software Gen5, both from BioTek Instruments GmbH (Bad Friedrichshall, Germany).

Measurements of IL-6, PGE_2_ and MMP1 in the supernatant: IL-6 human uncoated ELISA Kit (#88-7066-22, Invitrogen), Prostaglandin E_2_ ELISA Kit (DRG Instruments, Marburg, Germany, #EIA-5811) and RayBio human MMP1 ELISA Kit (RayBiotech Inc., Norcross, GA, USA, #ELH-MMP1-5). The absorptions were analyzed at 450 nm for all three ELISA Kits using the microplate spectrophotometer Epoch and the software Gen5.

Measurement of GAG: The amount of GAG was measured spectrophotometrically using 1,9-dimethylmethylene blue (DMB, Sigma-Aldrich, Taufkirchen, Germany). A standard curve of bovine chondroitin sulfate (Sigma-Aldrich) was generated to calculate the GAG concentration. The absorption was measured at 525 nm using the microplate spectrophotometer Epoch and the software Gen5.

### 2.5. Vitality of Chondrocytes

Chondrocytes were plated on 96-well culture plates at a density of 4 × 10^4^ cells/cm^2^. The cells were cultured and stimulated as described above. For testing the impact of mediators used on the viability of chondrocytes, the LIVE/DEAD Viability/Cytotoxicity Kit from Invitrogen (#L3224) was performed according to the manufacturer’s instructions. The assay determines the percentage of living and dead cells after stimulation with different mediators. The fluorescence intensity was measured using the microplate reader CLARIOstar (BMG LABTECH GmbH, De Meern, The Netherlands).

### 2.6. Measurement of Mitochondrial Function and Glycolysis

Functional parameters of mitochondrial and non-mitochondrial respiration and glycolysis were calculated based on the oxygen consumption rate (OCR) and the extracellular acidification rate (ECAR) measured with the Seahorse XF Cell Mito Stress Test Kit using the Seahorse XF Analyzer (Agilent, Santa Clara, CA, USA). The assay includes modulators of the mitochondrial respiration chain to determine mitochondrial ATP-linked respiration, maximal respiration, spare respiratory capacity, proton leak and non-mitochondrial respiration. These functional parameters were calculated by measuring the OCR of the cultured chondrocytes using the data analysis tool Seahorse XF Report Generator (Agilent). Oligomycin, carbonyl cyanide-4 (trifluoromethoxy) phenylhydrazone (FCCP) and rotenone/antimycin A were sequentially applied to the cultured chondrocytes to modulate the mitochondrial respiration. The basal OCR rate measured before application of these modulators consists of mitochondrial ATP-linked OCR, non-mitochondrial OCR by other cellular oxidative processes and mitochondrial OCR not coupled to ATP production, where protons return to the mitochondrial matrix independently of ATP synthase, which is called proton leak. The mitochondrial ATP production was calculated by decreased OCR after injection of oligomycin. The spare respiratory capacity was calculated from the maximal OCR after injection of the uncoupler FCCP, where the respiratory chain operates at maximum capacity, minus mitochondrial ATP-linked OCR, proton-leak-linked OCR and non-mitochondrial OCR. Finally, non-mitochondrial respiration was measured after application of the inhibitors rotenone and antimycin A. The glycolytic activity was indicated by the ECAR measured directly before application of oligomycin. The ECAR is mainly generated by lactate production through glycolysis and reflects the glycolytic activity, but a small amount of H^+^ can also be produced by non-glycolytic acidification.

For this purpose, chondrocytes were plated on Seahorse cell culture plates at a density of 3 × 10^4^ cells/well and cultured in chondrocyte culture medium for 6 days at 37 °C. Thereafter cells were stimulated with 0.1 ng/mL IL-1β or 0.1 µg/mL TNF for 24 h. Additionally, cells were stimulated with IL-1β or TNF, in combination with Forskolin (50 µM), and the opioids β-endorphin (1 µg/mL), nociceptin (1 µg/mL) or dynorphin A (1 µg/mL), for 24 h. The cells were plated with 8 repetitions for each combination of stimulation. Experiments were performed with a minimum of 5 biological replicates (donors), except stimulation experiments with opioid peptides on non-OA chondrocytes. The number of isolated hip chondrocytes from one donor was not enough to complete all experiments. Therefore, the number of donors varied between the experiments. Fewer patients with hip fractures and without any sign of osteoarthritic alterations of the cartilage (non-OA group) could be included in the study.

### 2.7. Statistical Analysis

For statistical analyses, the software SPSS statistics 21 (SPSS, Inc, Chicago, IL, USA) was used. Results are expressed as means ± SD or percent of control or displayed as box plots. Box plots display the median, the 25th percentile and the 75th percentile; whiskers represent the minimal and maximal values. Unstimulated chondrocytes were defined as control group. To compare non-OA and OA chondrocytes or unstimulated cells (control) and stimulated cells, the Mann–Whitney *U* Test was used. The Kruskal–Wallis test followed by Mann–Whitney *U* test and Bonferroni correction for multiple comparisons were used to evaluate the effects of inflammatory stimuli, forskolin or opioid neuropeptides on the measured metabolic parameters. Significance was set at *p* < 0.05.

## 3. Results

### 3.1. Comparison of Metabolic Processes of Unstimulated Non-OA and OA Chondrocytes

To evaluate whether there are permanent basal metabolic alterations induced by OA, the basal release of NO, IL-6, PGE_2_ and MMP1 was measured (Figure 1). The release of NO was similar from non-OA and OA chondrocytes. In contrast, the basal release of IL-6, MMP1 and PGE_2_ was significantly higher from OA chondrocytes compared to non-OA chondrocytes. The production of GAG was significantly lower from OA compared to non-OA chondrocytes.

Cellular oxidative processes can be determined by measuring the oxygen consumption rate in combination with modulators of oxidative processes (Figure 2a). The basal OCR, which includes the total oxygen consumption of all mitochondrial and non-mitochondrial oxidative processes, was not significantly different between unstimulated non-OA and OA chondrocytes (Figure 2). Furthermore, the mitochondrial ATP production was not significantly different between unstimulated non-OA and OA chondrocytes (Figure 2b,c). In contrast, unstimulated non-OA chondrocytes showed a significantly lower rate of non-mitochondrial respiration compared to OA chondrocytes. The glycolytic rate based on the ECAR was not significantly different between unstimulated non-OA and OA chondrocytes (Figure 2d).

### 3.2. Comparison of Metabolic Processes of Cytokine-Stimulated Non-OA and OA Chondrocytes

After stimulation with IL-1β or TNF, the basal OCR was comparable between non-OA and OA chondrocytes, independent of stimulation (Figure 2b,c). The mitochondrial ATP production, which was comparable between unstimulated non-OA and OA chondrocytes, was significantly lower in non-OA chondrocytes after stimulation with IL-1β or TNF. After cytokine stimulation, the mitochondrial ATP production decreased, on average, to 17.9% in non-OA chondrocytes and to 48.1% in OA chondrocytes after stimulation with IL-1β, and to 33.4% in non-OA chondrocytes and to 94.5% in OA chondrocytes after stimulation with TNF. The spare respiration capacity was higher in OA chondrocytes. The non-mitochondrial respiration was significantly higher in unstimulated OA chondrocytes compared to non-OA chondrocytes, but after stimulation with cytokines, the non-mitochondrial respiration was similar. Thus, the non-mitochondrial respiration increased, on average, 3-fold in non-OA chondrocytes compared to 1.4-fold in OA chondrocytes after stimulation with IL-1β, and 2-fold in non-OA chondrocytes compared to 1.2-fold in OA chondrocytes after stimulation with TNF. The ECAR was not significantly different between non-OA and OA chondrocytes, independent of cytokine stimulation (Figure 2d).

In summary, non-OA chondrocytes responded to inflammatory stimuli much more strongly compared to OA chondrocytes, expressed by a larger reduction in mitochondrial ATP production and a greater increase in non-mitochondrial oxygen consumption. The larger reduction in mitochondrial ATP production was not compensated by a higher cytokine-induced shift to glycolytic activity in non-OA chondrocytes.

### 3.3. Effect of Inflammatory Mediators and cAMP on Oxygen Consumption Rate and ECAR

As pointed out in the introduction, elevation of the cAMP level attenuates mitochondrial dysfunction in OA chondrocytes induced by IL-1β. To test whether cAMP affects IL-1β-induced or TNF-induced changes in the oxygen consumption rates and the different calculated mitochondrial parameters in both OA and non-OA chondrocytes, the stimulator of adenylyl cyclase forskolin was applied, together with IL-1β or TNF. The basal OCR did not change significantly after IL-1β or IL-1β + forskolin stimulation in either non-OA chondrocytes or in OA-chondrocytes (Figure 3). In non-OA chondrocytes, the ATP production and spare respiratory capacity both decreased considerably after IL-1β stimulation. This reduction was not significantly modified by forskolin co-application. In contrast, in OA chondrocytes, the reduction of ATP production after IL-1β stimulation was prevented by simultaneous application with forskolin. In non-OA chondrocytes, non-mitochondrial oxygen consumption increased significantly after IL-1β stimulation compared to unstimulated cells, whereas in OA chondrocytes, the increase was weaker and not significant. Co-application of IL-1β with forskolin had no significant effects on the IL-1β-induced increase in non-mitochondrial respiration in non-OA chondrocytes and OA chondrocytes. After stimulation with TNF, the mitochondrial ATP production and the spare respiratory capacity decreased significantly in non-OA chondrocytes, and this did not change significantly after co-application with forskolin. In contrast, we could not measure a significant effect of TNF or TNF + forskolin on OA chondrocytes. Application of forskolin alone did not affect the ATP production and non-mitochondrial respiration of non-OA and OA chondrocytes (Figure 3c). Application of DMSO (1:200, vehicle control) had no effect on these processes (Figure 3c).

In summary, forskolin reduced IL-1β-induced alterations in mitochondrial function in OA chondrocytes, whereas in non-OA chondrocytes, IL-1β induced stronger mitochondrial alterations which could not be prevented by co-application with forskolin. In contrast, forskolin had no significant effect on the IL-1β-induced increase in non-mitochondrial oxygen consumption in non-OA and OA chondrocytes. Application of TNF induced a significant reduction in mitochondrial ATP production and spare respiratory capacity in non-OA chondrocytes but not in OA chondrocytes.

The glycolytic rate significantly increased after stimulation with IL-1β in non-OA and OA chondrocytes but not after stimulation with TNF (Figure 4a,b). Co-application of IL-1β + forskolin or TNF + forskolin did not change the ECAR compared to IL-1β or TNF alone. Stimulation with forskolin alone significantly increased the ECAR in non-OA chondrocytes but not in OA chondrocytes (Figure 4c, DMSO control had no effect).

In summary, application of forskolin increased the glycolytic rate only in non-OA chondrocytes, but it had no additional effect on the IL-1β-induced increase in the glycolytic rate.

### 3.4. Cytokine-Induced Release of Pro-Inflammatory Mediators and Production of Glycosaminoglycan

Following stimulation with IL-1β or TNF, the release of NO, MMP1, PGE_2_ and IL-6 significantly increased in non-OA and OA chondrocytes (control vs. IL-1β/TNF-stimulated cells, all *p* < 0.005; Mann–Whitney *U* test; Figure 5). After stimulation with IL-1β and TNF, the release of NO was significantly higher in non-OA compared to OA chondrocytes (Figure 5a). The release of MMP1 after IL-1β and TNF stimulation was comparable in non-OA and OA chondrocytes (Figure 5b). In contrast, the IL-1β- and TNF-induced release of PGE_2_ was significantly higher in OA chondrocytes, independent of stimulation (Figure 5c). IL-1β induced a large increase in IL-6 release without difference between non-OA and OA, whereas the TNF-induced release of IL-6 was significantly higher in OA compared to non-OA chondrocytes (Figure 5d).

The production of GAG was significantly lower in OA compared to non-OA chondrocytes, with and without stimulation of pro-inflammatory mediators (Figure 5e). IL-1β and TNF significantly decreased GAG production in non-OA chondrocytes (IL-1β: *p* = 0.003, TNF: *p* = 0.016; Mann–Whitney *U* test), whereas in OA chondrocytes, the GAG production was not significantly affected by IL-1β or TNF.

### 3.5. Impact of Mediators on Vitality of Chondrocytes

To control whether IL-1β-induced or TNF-induced metabolic alterations were based on cytotoxic effects, an assay was performed to determine the percentage of living and dead chondrocytes after stimulation with IL-1β and TNF compared to unstimulated cells. The cytotoxic dimethyl sulfoxide (DMSO) control (1:10) proved the validity of the test. The percentages of living cells and dead cells did not change significantly after stimulation with IL-1β or TNF in either non-OA or OA chondrocytes (Figure 6).

### 3.6. Impact of Opioid Neuropeptides on NO Release and Oxygen Consumption Rate

To evaluate whether opioid neuropeptides have a significant effect on IL-1β- or TNF-induced metabolic alterations, non-OA and OA human hip chondrocytes were stimulated with IL-1β or TNF, in combination with either β-endorphin, nociceptin or dynorphin A. There were no significant effects of β-endorphin, nociceptin and dynorphin A on IL-1β- or TNF-induced NO-release (Figure 7a,b) or on mitochondrial and non-mitochondrial oxygen consumption rates (Figure 7c–e) or on glycolytic activity (Figure 8a,b).

## 4. Discussion

Our study compared critical metabolic processes of non-OA and OA human chondrocytes to evaluate whether OA initiates permanent metabolic alterations and whether non-OA and OA chondrocytes respond similarly to inflammatory stimuli. The results of the current study provide evidence that OA induces metabolic alterations in human hip chondrocytes, expressed by a higher basal release of inflammatory mediators and increased non-mitochondrial oxidative processes in OA chondrocytes. Surprisingly, the mitochondrial ATP production and glycolytic activity were comparable between unstimulated non-OA and OA hip chondrocytes. Non-OA chondrocytes responded much more strongly to inflammatory stimuli compared to OA chondrocytes. After stimulation with IL-1β or TNF, the release of NO was significantly higher and the reduction in mitochondrial ATP production was greater in non-OA chondrocytes, and this was not compensated by a stronger shift to glycolysis. Induction of cAMP production by the application of forskolin did not prevent IL-1β-induced mitochondrial dysfunction in non-OA chondrocytes but did so in OA chondrocytes. Activation of the opioid receptors interfering with the cAMP pathway did not alter mitochondrial function under inflammatory conditions.

A direct comparison of non-OA and OA chondrocytes is necessary to evaluate whether the alterations are only induced by experimental inflammatory stimuli or are permanently present beforehand and affect the metabolism. Mitochondrial dysfunction and a shift of oxidative phosphorylation to glycolysis has been reported in OA chondrocytes [16,17]. Reduced activity in mitochondrial complexes II and III was found in OA chondrocytes [18]. Furthermore, decreased mitochondrial ATP production and mitochondrial dysfunction were described in chondrocytes after stimulation with pro-inflammatory mediators, mechanical overload and the induction of reactive oxygen species [8,9,19]. Our results reveal that the basal mitochondrial ATP production and the basal glycolytic activity are comparable between unstimulated non-OA and OA hip chondrocytes.

A detailed proteomics analysis of non-OA and OA chondrocytes described 23 differently expressed mitochondrial proteins [20]. The study confirmed the decreased expression of mitochondrial complexes II and III as well as the decreased presence of SOD2, which plays an important anti-oxidative role. However, the study also reported an increased abundance of two subunits of complex I and an increased abundance of TRAP1, which is located in the mitochondria and antagonizes ROS, protecting chondrocytes from oxidative stress-induced apoptosis [20]. The mitochondrial mass was increased in OA chondrocytes, which could compensate for the deficiencies in complexes II and III [18]. Thus, different alterations may compensate for each other and allow normal basal mitochondrial ATP production in OA chondrocytes, as observed in the present study.

Specific adaptation mechanisms to oxidative stress in OA chondrocytes may also provide an explanation for the reduced sensitivity of OA chondrocytes to stimulation with IL-1β or TNF (less NO production and less reduction in mitochondrial ATP production in OA chondrocytes than in non-OA chondrocytes). Our previous study revealed a direct connection between IL-1β-induced NO production and the IL-1β-induced mitochondrial dysfunction in human knee OA chondrocytes [5]. Blocking iNOS activity prevented NO production and IL-1β-induced mitochondrial alterations. In the present study, the basal release of NO was very low and comparable between non-OA and OA chondrocytes, but after stimulation with IL-1β or TNF, the release of NO was significantly higher in non-OA chondrocytes. Thus, a higher production of NO in non-OA chondrocytes could explain the greater reduction in mitochondrial ATP production. Since NO production by iNOS consumes oxygen, the increased non-mitochondrial oxygen consumption in non-OA chondrocytes could reflect this oxidative process. Our interpretation is, therefore, that non-OA chondrocytes react more sensitively to inflammatory stimuli, resulting in a higher production of NO, which then inhibits the ATP production of the mitochondria. Since inhibition of iNOS restored ATP production [5], the NO-induced inhibition of mitochondrial ATP production may be reversible.

Why non-OA chondrocytes produced more NO compared to OA chondrocytes after cytokine stimulation is unclear. There must be processes reducing iNOS expression or activity in OA chondrocytes. We previously reported that cAMP or cAMP-inducing molecules such as forskolin or PGE_2_ reduce NO production, preventing mitochondrial dysfunction in chondrocytes [5]. We confirmed, in human hip OA chondrocytes, the preventive effect of cAMP on IL-1β-induced mitochondrial alterations. Now we could show that the production of PGE_2_ was significantly higher in OA chondrocytes, which could be the reason for the reduced NO production. This finding indicates protective functions of PGE_2_. Since increased cAMP induced by forskolin did not prevent IL-1β effects on mitochondria in non-OA hip chondrocytes, this cAMP-associated mechanism of reducing IL-1β effects is only active in OA chondrocytes. Because different molecules can induce cAMP, this is an interesting mechanism targeting OA processes, whereas processes in non-OA cells are not affected.

Since chondrocytes utilize mainly glycolysis for ATP production, the importance of oxidative phosphorylation for total cellular ATP production cannot be assessed. In normal chondrocytes, aerobic and anaerobic glycolysis coexist under physiological oxygen tension [17]. Only up to 25% of the ATP is produced by mitochondrial oxidative phosphorylation in chondrocytes. Because mitochondrial dysfunction can induce apoptosis, oxidative stress and an increased production of inflammatory mediators, intact mitochondrial function is crucial for the homeostasis and survival of chondrocytes. In normal human chondrocytes, inflammatory stimuli such as IL-1β or TNF, in combination with mitochondrial respiratory inhibitors inducing mitochondrial dysfunction, increase the expression and release of IL-8 or PGE_2_ [21]. In addition to the reduction in mitochondrial ATP production, IL-1β or TNF induce a shift to higher glycolytic activity [17]. Our results indicate such a shift of oxidative phosphorylation to glycolysis after cytokine stimulation in both non-OA and OA human hip chondrocytes. However, basal and cytokine-induced glycolytic activity were similar between non-OA and OA chondrocytes, and therefore, the greater reduction in mitochondrial ATP production in non-OA chondrocytes was not compensated by a greater increase in glycolytic activity.

The significantly higher basal release of the pro-inflammatory mediators IL-6 and PGE_2_ in OA chondrocytes, and the stronger release after stimulation with IL-1β or TNF, may contribute to OA pain. IL-6 is an important pain mediator, which sensitizes nociceptive neurons to mechanical stimuli [22,23]. The level of IL-6 in the synovial fluid is associated with OA pain in patients with knee OA as well as hip OA [24,25]. In traumatic injured knees, the level of IL-6 in the synovial fluid is significantly lower compared to OA knee joints [26]. The measured high basal release of IL-6 by OA chondrocytes could provide the high IL-6 concentrations in the synovial fluid, which could be the reason for chronic pain conditions in OA-affected joints.

As expected, the production of the matrix protein glycosaminoglycan was higher and the release of the matrix degenerative enzyme MMP1 was lower in non-OA chondrocytes.

Moreover, several neuropeptides were detected in the synovial fluid. The effects of these neuropeptides on the metabolism of chondrocytes can differ in non-OA and OA human knee chondrocytes [10]. Endogenous opioids were also detected in the synovial fluid of human hip and knee OA [27]. Several cell types in the synovial tissue can produce opioids and their receptors, especially immune cells and sensory nerve fibers [28]. Inflammation induces upregulation of both, opioids and opioid receptors, counteracting pain and inflammation. Chondrocytes also express opioid receptors, which are G-protein-coupled receptors affecting the cAMP pathway [11]. Interestingly, activation of opioid receptors by the opioid β-endorphin decreases cAMP signaling [11], whereas the opioid dynorphin A increases cAMP signaling in chondrocytes [14]. Our experiments do not indicate an effect of the opioids β-endorphin, dynorphin A and nociceptin on the IL-1β- and TNF-induced NO production or on cytokine-induced mitochondrial dysfunction in non-OA and OA hip chondrocytes. Opioid receptors are a promising target for the development of therapeutics for OA patients, since opioids are anti-inflammatory and analgesic molecules and the activation of some opioid receptors revealed protective effects against cartilage degeneration and injury [14]. A phase 2 clinical trial (NCT02944448) demonstrated a significant pain reduction after application of a selective opioid-receptor agonist in hip OA patients compared to placebo patients [29]. However, opioids can also induce the expression of pro-inflammatory mediators in chondrocytes [15]. Since local application of opioids is a promising option for pain treatment in OA, it is important that our study did not demonstrate a negative impact of β-endorphin, dynorphin A and nociceptin on mitochondrial function under inflammatory conditions.

Targeting mitochondrial pathways by small molecules to improve mitochondrial function in OA chondrocytes is a new strategy for OA treatment [16]. Several studies addressed the application of exogenous drugs, such as antioxidants, drugs to inhibit the mitochondrial apoptotic pathway or to improve mitochondrial function [16]. Additional research is necessary to analyze whether improving mitochondrial function with these small molecules also has an effect on pain relief in OA patients.

## 5. Conclusions

The direct comparison of non-OA and OA chondrocytes presents various OA-induced metabolic alterations in human hip chondrocytes, and some of them appeared only after stimulation with pro-inflammatory mediators. The high basal release of IL-6 from OA chondrocytes could be responsible for chronic pain under OA conditions. In contrast to our initial hypothesis, non-OA chondrocytes respond much more strongly to inflammatory stimuli, with a large reduction in mitochondrial ATP production and increased NO production. The blunted response of OA chondrocytes to such stimuli suggests an upregulation of counteracting mechanisms, and one of them may be the upregulation of cAMP, possibly by PGE_2_. Since opioids do not have a negative effect on the mitochondrial function of chondrocytes, their intra-articular application does not carry the risk of mitochondrial dysfunction.

## Figures and Tables

**Figure 1 biomedicines-10-01349-f001:**
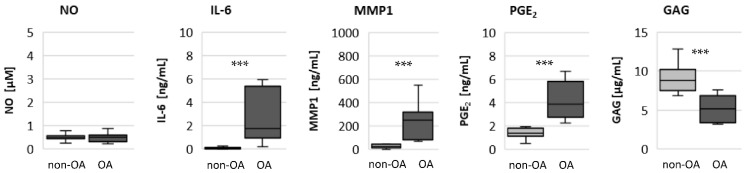
Basal release of NO, IL-6, MMP1, PGE_2_ and GAG from non-OA (number of patients (N) = 8) and OA (N = 10) human hip chondrocytes. Statistics: non-OA vs. OA chondrocytes, Mann–Whitney *U* Test. *** *p* < 0.005.

**Figure 2 biomedicines-10-01349-f002:**
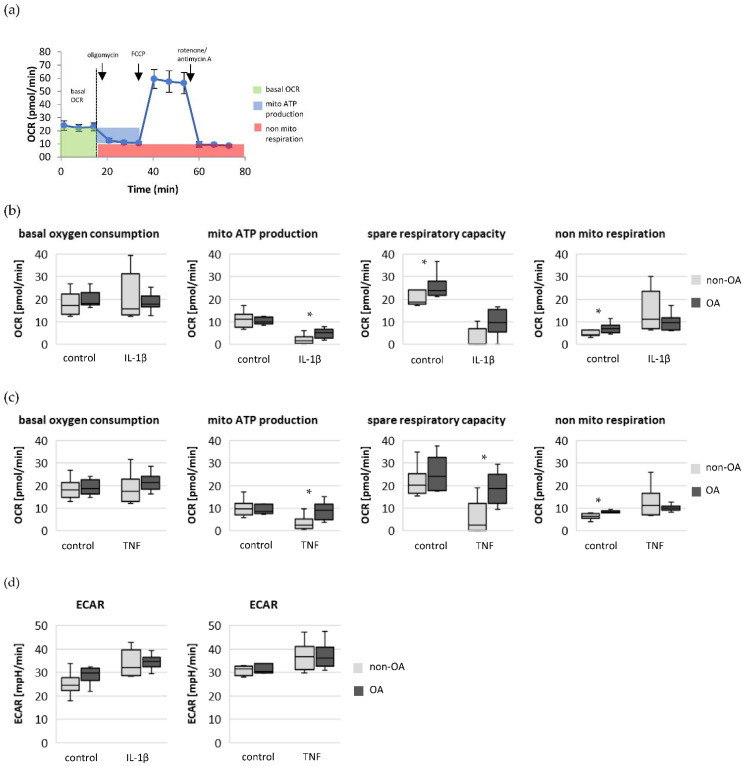
Basal oxygen consumption rate, mitochondrial ATP production, spare respiratory capacity, non-mitochondrial respiration and ECAR of non-OA and OA chondrocytes. (**a**) Schema of the OCR measurement with application of oligomycin, FCCP and rotenone + antimycin A. (**b**) OCR values of unstimulated (control) and IL-1β-stimulated cells (non-OA patients = 6, OA patients = 7). (**c**) OCR values of unstimulated (control) and TNF-stimulated cells (non-OA patients = 6, OA patients = 6). (**d**) Comparison of ECAR values of unstimulated (control), IL-1β-stimulated (non-OA patients = 6, OA patients = 7) and TNF-stimulated cells (non-OA patients = 6, OA patients = 6). Statistics: non-OA vs. OA chondrocytes, Mann–Whitney *U*-test. Values are displayed as box plots. * *p* < 0.05. OCR: oxygen consumption rate; mito: mitochondrial; ECAR: extracellular acidification rate.

**Figure 3 biomedicines-10-01349-f003:**
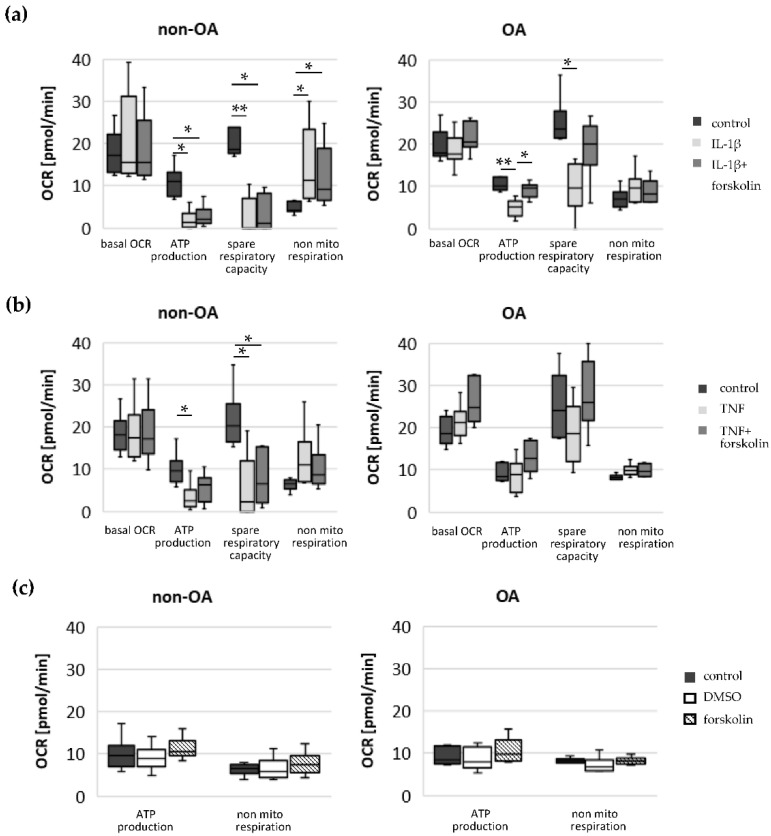
Changes in basal oxygen consumption rate, mitochondrial ATP production, spare respiratory capacity and non-mitochondrial respiration of non-OA and OA chondrocytes before and after stimulation with (**a**) IL-1β and forskolin (non-OA patients = 6, OA patients = 7), (**b**) TNF and forskolin (non-OA patients = 6, OA patients = 6) and (**c**) forskolin and DMSO (non-OA patients = 6, OA patients = 6). Values are displayed as box plots. Statistics: Kruskal–Wallis test followed by Mann–Whitney *U* test and Bonferroni correction. ** *p* < 0.01 and * *p* < 0.05. Control: unstimulated cells; DMSO: dimethyl sulfoxide; OCR: oxygen consumption rate; mito: mitochondrial.

**Figure 4 biomedicines-10-01349-f004:**
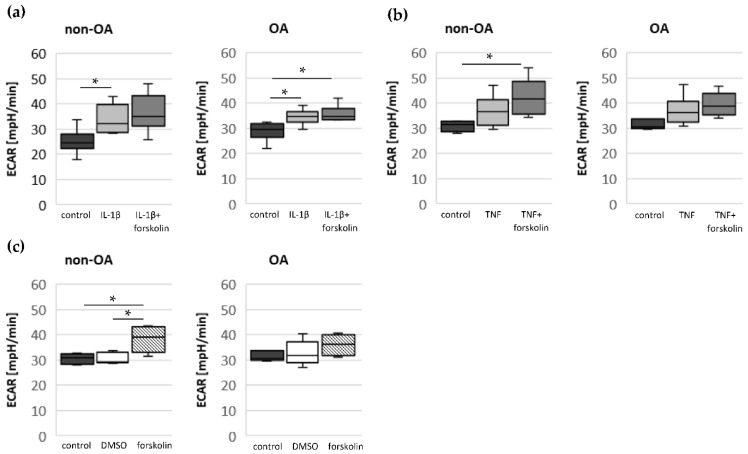
Glycolytic rate measured by ECAR in unstimulated cells (control) and after stimulation with (**a**) IL-1β and forskolin (non-OA patients = 6, OA patients = 7), (**b**) TNF and forskolin (non-OA patients = 6, OA patients = 6) and (**c**) forskolin and DMSO (non-OA patients = 6, OA patients = 6). Values are displayed as box plots. Statistics: Kruskal–Wallis test followed by Mann–Whitney *U* test and Bonferroni correction. * *p* < 0.05. DMSO: dimethyl sulfoxide; ECAR: extracellular acidification rate.

**Figure 5 biomedicines-10-01349-f005:**
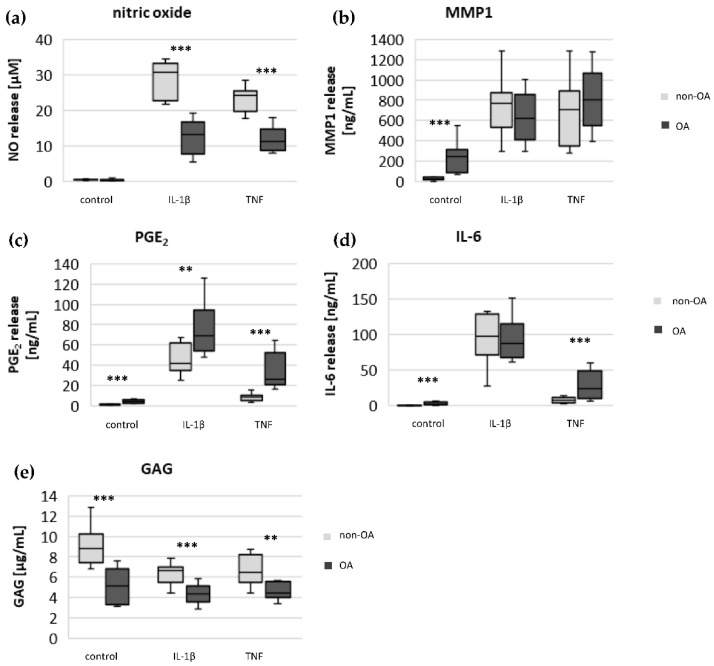
Impact of IL-1β and TNF on the release of NO (**a**), MMP1 (**b**), PGE_2_ (**c**), IL-6 (**d**) and GAG (**e**) in non-OA (number of patients = 8) and OA (number of patients = 10) human hip chondrocytes. Values are displayed as box plots. Statistics: non-OA vs. OA chondrocytes, Mann–Whitney *U* Test, *** *p* < 0.005, ** *p* < 0.01. Control: unstimulated cells.

**Figure 6 biomedicines-10-01349-f006:**
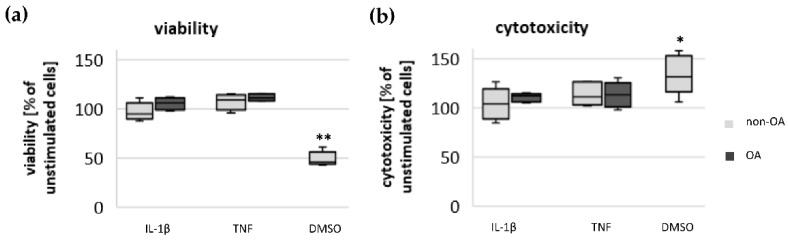
Impact of IL-1β and TNF on (**a**) viability and (**b**) cytotoxicity of non-OA (number of patients = 6) and OA (number of patients = 4) human hip chondrocytes. Stimulation with dimethyl sulfoxide (DMSO 1:10, number of patients = 5) proved the validity of the tests. Values are normalized to unstimulated cells and displayed as box plots. Statistics: non-OA/OA/DMSO vs. unstimulated cells, Mann–Whitney *U* test; no significant differences were found between IL-1β-/TNF-stimulated vs. unstimulated chondrocytes. ** *p* < 0.01 and * *p* < 0.05.

**Figure 7 biomedicines-10-01349-f007:**
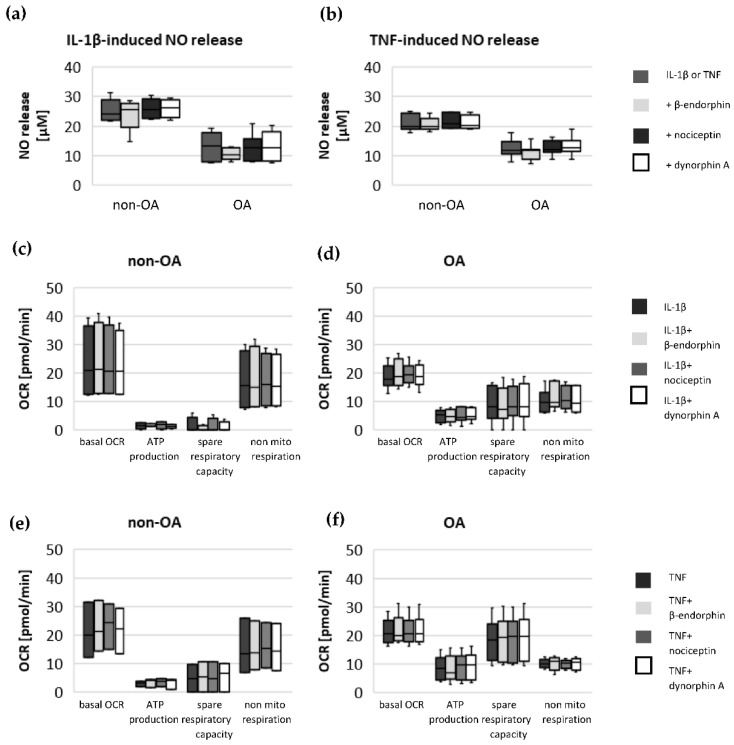
Impact of opioid neuropeptides on the (**a**) IL-1β-induced and (**b**) TNF-induced release of NO (non-OA patients = 5, OA patients = 8) and on the (**c**,**d**) IL-1β-induced (non-OA patients = 4, OA patients = 6) and (**e**,**f**) TNF-induced (non-OA patients = 3, OA patients = 5) changes in basal oxygen consumption rate, mitochondrial ATP production, spare respiratory capacity and non-mitochondrial respiration in non-OA and OA chondrocytes. Values are displayed as box plots. Statistics: Kruskal–Wallis test; no significant differences were found within the groups.

**Figure 8 biomedicines-10-01349-f008:**
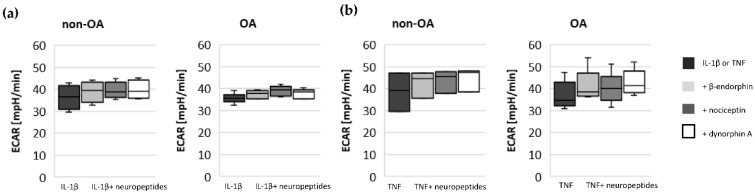
Impact of opioid neuropeptides on glycolytic activity measured by ECAR after stimulation with (**a**) IL-1β and opioid neuropeptides (non-OA patients = 4, OA patients = 6) and (**b**) TNF and opioid neuropeptides (non-OA patients = 3, OA patients = 5). Values are displayed as box plots. Statistics: Kruskal–Wallis test; no significant differences were found within the groups. ECAR: extracellular acidification rate.

## Data Availability

All original data of the study are presented in the article and queries can be directed to the corresponding author.

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
