# Peer review of "Osteoarthritis-Induced Metabolic Alterations of Human Hip Chondrocytes"

_biomedicines, 2022, doi:10.3390/biomedicines10061349_

Round 1

Reviewer 1 Report

The metabolism of chondrocytes in osteoarthritis has always been the focus of research and treatment of the disease, in which mitochondrial biological properties are the basis of cell metabolism. By comparing the differences between mitochondrial ATP production and basal oxygen consumption rates stimulated by IL-1β, TNF, Forskolin and opioid peptides under OA and non-OA, this study preliminarily elucidated the changes of chondrocyte metabolism under OA, which is scientifically sound.

The found of which non-OA chondrocytes respond much stronger to inflammatory stimuli than OA chondrocytes is certainly interesting. The study has convincingly demonstrated that OA chondrocytes to such stimuli suggests an upregulation of counteracting mechanisms, and one of them may be the upregulation of cAMP possibly by PGE.

However, it is suggested that if other positive data can be provided, especially for reagent related to OA treatment, it will be more novel and convincing. In order to keep up with the times, it is recommended to search for reagents mentioned above in recent research and discuss in the discussion part.

Author Response

We are pleased about your positive response and we thank you for your constructive comments.

We included new treatment options in the discussion section on page 14.

Selective opioid receptor agonists are potential reagents for pain treatment in OA patients. A phase 2 clinical trial (NCT02944448) demonstrated a significant pain reduction after application of a selective opioid receptor agonist in hip OA patients compared to placebo patients. Thus, the result of our study, that local application of opioids has no negative impact on mitochondrial function under inflammatory conditions, is important for the development of this treatment option. Additionally, targeting mitochondrial pathways by small molecules to improve mitochondrial function in OA chondrocytes is a new strategy for OA treatment. Thus, analyses of basal mechanisms of mitochondrial processes are important. Restoring the normal mitochondrial function in OA chondrocytes by potential drugs addressing mitochondrial pathways will be a key aspect in our future experiments.

Reviewer 2 Report

In this work, metabolic changes in primary chondrocytes from patients with osteoarthritis were evaluated.  The authors found that OA chondrocytes expresses higher levels of IL-6 and PGE2, but they show OCRs similar to those of the non-OA chondrocytes. Non-OA chondrocytes show greater responses to Il6 and TNF compared with OA chondrocytes.

Major:

  1. This paper is descriptive in nature.  The lack of mechanistic data makes the finding less convincing and the significance unclear.  For example, the authors show no experimental data to explain the differences between non-OA and OA cells, like why mitochondrial activity is low in on-OA, why non-OA cells respond to inflammatory stimuli strongly, etc.  It appears that some of such questions can be addressed by relatively straightforward and easy experiments, such as assessments of glycolysis, activation of intracellular signaling upon IL/TNF treatment, etc. 

Other specific comments:

  1. Table 1. These data would be better presented in plots with individual data point.  Since these data are from cells from different individuals, SD instead of SE should be used.

  1. How did the authors evaluated the quality of cultured chondrocytes? It is often observed that primary chondrocytes lose their characteristics after in vitro culture.

  1. Fig. 1. The authors should state the interpretation of the OCR data.  Non-OA cells appear to show lower ETC/ATP synthase activity than OA cells.  Is it because of reduced amounts of mitochondria, or increased shift to glycolysis, or reduced TCA cycle metabolism?  At least, the authors should show ECAR data to interpret the differences in bioenergetics.

  1. OCR should be corrected to cell numbers (i.e. pmol/min/cells). The cell number should be counted when the analysis is performed.

  1. Fig. 1. And 2. The pie chart data should be compared between non-OA and OA samples with statistical analysis, like chi-square test.

  1. Fig.2. How do the authors interpret the large fraction of non-mitochondria-dependent O2 consumption? Based on the data in Fig.4, it looks like that IL1b/TNF treatment substantially suppress the mitochondrial activity; so it is not an increase in non-mitochondrial O2 consumption (or just simply a non-specific background) but is a decrease in mitochondrial O2 consumption. 

  1. Fig. 4. These data suggest that Il1/Tnf suppresses mitochondrial activity. What happens to ECAR? what happens to other bioenergetic markers, such as p-AMPK, lactate, NAD, etc upon IL/TNF treatment?

  1. Fig. 6. Vehicle only control is missing.

Author Response

We are pleased about your constructive comments. The next paragraphs contain our response to your comments and indicate which changes have been made in the revised manuscript.

We included ECAR data to present the glycolytic activity of non-OA and OA chondrocytes in Figure 2, 4 and 8 to improve the manuscript, as recommended.

Other specific comments: 

  1. Table 1. These data would be better presented in plots with individual data point.  Since these data are from cells from different individuals, SD instead of SE should be used. 

The data of the basal release of NO, IL-6, MMP1, PGE2 and GAG (previously Table 1) are now presented in Figure 1 (page 5) as Boxplots, as recommended.

  1. How did the authors evaluated the quality of cultured chondrocytes? It is often observed that primary chondrocytes lose their characteristics after in vitro culture. 

After the isolation procedure, the chondrocytes were directly seeded into 24-well culture plates, and were cultivated only for 5 days before stimulation. In many other studies, chondrocytes are subcultured for up to 21 days to reach confluence, before starting experiments, and chondrocytes are used after some passages to increase the cell number and optimize the culture conditions. Under these conditions chondrocytes start to change gene-expression. The short cultivation of 5 days only reduces the risk of de-differentiation but we had to accept to obtain a lower cell number per patient and to perform fewer experiments. The basal release of mediators before stimulation (Figure 1) from OA and non-OA chondrocytes showed clear differences suggesting the basal properties of chondrocytes are not distorted by cultivation for 5 days.

  1. Fig. 1. The authors should state the interpretation of the OCR data.  Non-OA cells appear to show lower ETC/ATP synthase activity than OA cells.  Is it because of reduced amounts of mitochondria, or increased shift to glycolysis, or reduced TCA cycle metabolism?  At least, the authors should show ECAR data to interpret the differences in bioenergetics. 

We thank the reviewer for listing several mechanisms possibly underlying the changes observed. As suggested we now show the ECAR data (see Figures 2, 4 and 8). After stimulation with cytokines, the ECAR data showed an increase of glycolytic activity. However, the ECAR data were not significantly different between non-OA and OA chondrocytes, neither before nor after stimulation. Thus, we could observe a shift to glycolysis, but this shift was not greater in non-OA chondrocytes to compensate the greater reduction of mitochondrial ATP production after cytokine-stimulation. We believe that the higher NO production of non-OA chondrocytes could explain the lower mitochondrial ATP production compared to OA chondrocytes. Our previous study (Eitner et al, Int J Mol Sci. 2021, 22(5); DOI: 10.3390/ijms22052477) revealed, that blocking iNOS activity inhibits NO-induced reduction of mitochondrial activity. Since NO production by iNOS consumes oxygen, the increased non-mitochondrial oxygen consumption also indicates more oxidative processes in non-OA chondrocytes. Our interpretation of the OCR data is, therefore, that non-OA chondrocytes react more sensitive to inflammatory stimuli resulting in a higher production of NO, which inhibits ATP production of mitochondria. Since inhibition of iNOS restored ATP production, this NO-induced inhibition of mitochondrial ATP production could be reversible. We have addressed this point in the discussion section on page 13, lines 458-465.

  1. OCR should be corrected to cell numbers (i.e. pmol/min/cells). The cell number should be counted when the analysis is performed. 

We seeded the same cell number in every well. Thus, in all experiments the same cell number and the same culture conditions were used. Unfortunately, we observed some detachment of cells after application of rotenone and antimycin A at the end of the seahorse experiment. Thus, the calculation of the cell number after the Seahorse experiment may not reflect the real cell number during the Seahorse experiment, and therefore, we decided not to correct to cell number to avoid more variables.

  1. Fig. 1. And 2. The pie chart data should be compared between non-OA and OA samples with statistical analysis, like chi-square test. 

The pie chart data in the original manuscript displayed the percentage of specific processes of the total oxygen consumption. The real amounts of OCR were displayed in Figure 3 with the appropriate statistical analysis. Since this presentation caused confusion, we removed the pie charts and concentrate on the comparison of the real OCR values. The comparison of non-OA and OA OCR values is now displayed in the new Figure 2 on page 6.

  1. Fig.2. How do the authors interpret the large fraction of non-mitochondria-dependent O2 consumption? Based on the data in Fig.4, it looks like that IL1b/TNF treatment substantially suppress the mitochondrial activity; so it is not an increase in non-mitochondrial O2 consumption (or just simply a non-specific background) but is a decrease in mitochondrial O2 consumption.  

As mentioned above, we removed the old pie charts in Figures 1 and 2 and show now the comparison of the real OCR values in the new Figure 2. We observed a reduction of mitochondrial activity and an increase in non-mitochondrial oxygen consumption in non-OA chondrocytes compared to OA-chondrocytes. We believe, that both reactions are based on an increased production of NO.

  1. Fig. 4. These data suggest that Il1/Tnf suppresses mitochondrial activity. What happens to ECAR? what happens to other bioenergetic markers, such as p-AMPK, lactate, NAD, etc upon IL/TNF treatment? 

Now we present the ECAR data in Figure 4 on page 9. After stimulation with IL-1β we observed an increase of ECAR reflecting an increase of glycolytic activity in non-OA and OA-chondrocytes. The ECAR was not different between non-OA and OA chondrocytes. Thus, the stronger reduction of mitochondrial ATP production in non-OA chondrocytes was not compensated by a stronger increase of glycolytic activity.

  1. Fig. 6. Vehicle only control is missing.

The DMSO sample in Figure 6 was only to prove the validity of the test with a high concentration of DMSO and did not reflect a vehicle control. IL-1β, TNF and all opioid peptides were dissolved in water and were finally diluted in medium. Only forskolin was dissolved in 1:200 DMSO. Now we included data of vehicle control (DMSO 1:200) and application of forskolin alone in Figure 3 (OCR, mitochondrial ATP production and non-mitochondrial OCR) and in Figure 4 (ECAR) including all experiments using stimulation with forskolin. We included a statement in the method section, that IL-1β and TNF were dissolved in water (page 3, line 103.).

Reviewer 3 Report

Overall, this is a good work and novel. Need major revision for further consideration.

Specific comments:

Abstract

 stimulated with IL-1β, TNF, forskolin and opioid peptides.: Would be better, to briefly explain the reason to use this inducers

chondrocytes exhibited significantly higher basal release of IL-6, PGE2 and MMP1 ..than non-OA chondrocytes: Explain with or without inducers?

1. Introduction

Reduced activity of mitochondrial complexes have: Check Grammar

NO, IL-6, prostaglandin E2 (PGE2), MMP-3: Change to NO, IL-6, prostaglandin E2 (PGE2) and MMP-3

Whether the mitochondrial activity of normal chondrocytes is modulated by cAMP, has not been explored: Check, reframe

2. Materials and Methods

Isolation of human hip chondrocytes: Its better to confirm the purity of chondrocytes by flow cytometer.

Effective concentrations of all used mediators were determined in preliminary experiments: Its better to provide the data as supplementary file for reference.

TNF in combination with β-Endorphin (1 µg/mL), Nociceptin (1 µg/mL) or Dynorphin A (1 µg/mL). : How long?

ELISA Kit (#88-7066-22, Invitrogen), Prostaglandin E2 ELISA: change to ELISA Kit (#88-7066-22, Invitrogen), Prostaglandin E2 ELISA kit

The absorption was analyzed at 450 nm: Not clear. 450 nm for all three (IL-6, PGE2 and MMP1) test?

The amount of glycosaminoglycan (GAG): Always abbreviate at first usage.

Results:

It is recommended to analysis the chondrogenesis effect of these cytokines used in this study by Alcian Blue.

After stimulation with 0.1 ng/mL IL-1β for 24 hours,: Contradictory with methods (Refer Line no 132: for 48 hours)

non-OA and OA chondrocytes (control vs. IL-1β/TNF: Explain the control group details in method

4. Discussion

ATP production was comparable between...mitochondrial dysfunction in non-OA chondrocytes but in OA chondrocytes: Seems repeating the similar statement as results. try to provide a justification and mechanism with a concise description of the results.

Many studies investigated the effect ...are permanently present beforehand and affect the metabolism.: Looks like these statements suitable under Introduction part. Justifying the impact and necessary of this work. 

Need to refine the language (grammar) and typo-errors throughout the MS. Try to focus on writing the methodological details throughout, keep consistent procedure. 

Author Response

Overall, this is a good work and novel. Need major revision for further consideration.

We are pleased about your positive response and we thank you for your constructive comments. The next paragraphs contain our response to your comments and indicate which changes have been made in the revised manuscript.

Specific comments:

Abstract

 stimulated with IL-1β, TNF, forskolin and opioid peptides.: Would be better, to briefly explain the reason to use this inducers

Unfortunately, the abstract is limited to 200 words. Therefore, we explained the reason to use these inducers in the introduction.

chondrocytes exhibited significantly higher basal release of IL-6, PGE2 and MMP1 ..than non-OA chondrocytes: Explain with or without inducers?

We specified some sentences in the abstract.

  1. Introduction

Reduced activity of mitochondrial complexes have: Check Grammar

We have corrected this.

NO, IL-6, prostaglandin E2 (PGE2), MMP-3: Change to NO, IL-6, prostaglandin E2 (PGE2) and MMP-3

We have corrected this, as recommended.

Whether the mitochondrial activity of normal chondrocytes is modulated by cAMP, has not been explored: Check, reframe

We have corrected this.

  1. Materials and Methods

Isolation of human hip chondrocytes: Its better to confirm the purity of chondrocytes by flow cytometer.

Chondrocytes are the only cell type in the articular matured cartilage. We removed the cartilage carefully and washed the pieces of cartilage several time before isolation. To avoid de-differentiation, cells were cultivated only for 5 days and then used for the described experiments.

Effective concentrations of all used mediators were determined in preliminary experiments: Its better to provide the data as supplementary file for reference.

Preliminary experiments to determine the concentrations of used mediators are now presented in a supplementary file.

TNF in combination with β-Endorphin (1 µg/mL), Nociceptin (1 µg/mL) or Dynorphin A (1 µg/mL). : How long?

Cells were stimulated with IL-1β or TNF in combination with β-Endorphin (1 µg/mL), Nociceptin (1 µg/mL) or Dynorphin A (1 µg/mL) for 48 hours. Page 3, line 137.

ELISA Kit (#88-7066-22, Invitrogen), Prostaglandin E2 ELISA: change to ELISA Kit (#88-7066-22, Invitrogen), Prostaglandin E2 ELISA kit

We have corrected this, as recommended.

The absorption was analyzed at 450 nm: Not clear. 450 nm for all three (IL-6, PGE2 and MMP1) test?

We changed the sentence. Page 4, line 151.

The amount of glycosaminoglycan (GAG): Always abbreviate at first usage.

We changed this. Page 4, line 153.

Results:

It is recommended to analysis the chondrogenesis effect of these cytokines used in this study by Alcian Blue.

Alcian blue is often used to control the development of mesenchymal progenitors to chondrocytes. Since we stimulate matured chondrocytes for only 2 days, we did not expect chondrogenesis effects or chondrocyte differentiation.

After stimulation with 0.1 ng/mL IL-1β for 24 hours,: Contradictory with methods (Refer Line no 132: for 48 hours)

The release of NO, IL-6, PGE2, MMP1 were measured after stimulation with IL-1β or TNF after 48 hours. The measurements of OCR and ECAR with the seahorse analyzer were done after 24 hours.

non-OA and OA chondrocytes (control vs. IL-1β/TNF: Explain the control group details in method

Controls displayed values of unstimulated chondrocytes. We added a sentence to define the control group in 2.7. Statistical analysis on page 5 and define the control group in the figure legends.

  1. Discussion

ATP production was comparable between...mitochondrial dysfunction in non-OA chondrocytes but in OA chondrocytes: Seems repeating the similar statement as results. try to provide a justification and mechanism with a concise description of the results.

We think that the first paragraph of the discussion highlights the main results, and in many journals the first paragraphs is therefore a short summary. We would like to keep these sentences.

Many studies investigated the effect ...are permanently present beforehand and affect the metabolism.: Looks like these statements suitable under Introduction part. Justifying the impact and necessary of this work. 

We removed this sentence. We included some sentences in the discussion section on page 14 to underline the impact of our work.

Need to refine the language (grammar) and typo-errors throughout the MS. Try to focus on writing the methodological details throughout, keep consistent procedure. 

We revised the result section: we removed Figures 1 and 2, included new data presenting the glycolytic activity, and included data of vehicle control and stimulation with forskolin alone.

Reviewer 4 Report

"Osteoarthritis-induced metabolic alterations of human hip chondrocytes" by Eitner and colleagues is a very interesting work investigating the different persistence of OA and non-OA chondrocytes in mitochondrial metabolism and function, as well as the different responsiveness to stimulation with IL-1β, TNF and cAMP modulators. 
Overall, the paper is well structured and well written. The results are innovative and very interesting and undoubtedly provide new information in the field. I have only two suggestions for the authors. First, they should improve the layout and quality of the figures, especially Figures 1 and 2. Next, they should define all acronyms well throughout the text. 
After these minor revisions, it is likely that the manuscript will be accepted for publication.

Author Response

We are pleased about your positive response and we thank you for your constructive comments.

Since the pie charts caused confusion, we removed Figure 1 and 2 and concentrate on the comparison of the real OCR values. Additionally, we included data of stimulation with DMSO (vehicle control) and forskolin alone and ECAR values, which reflect the glycolytic activity of the chondrocyte.

Round 2

Reviewer 3 Report

The revised version is complete and satisfactory.